



# Nitrogen limitation information retrieved from data assimilation

Song Wang[1,2,3], Carlos A. Sierra[3], Yiqi Luo[4], Jinsong Wang[1,2], Weinan Chen[1,2], Yahai Zhang[3,5], Aizhong Ye[5], Shuli Niu[1,2]

[1]Key Laboratory of Ecosystem Network Observation and Modeling, Institute of Geographic Sciences and Natural Research, Chinese Academy of Sciences, Beijing 100101, China.

[2]College of Resources and Environment, University of Chinese Academy of Sciences, Beijing 100049, China.

[3]Department of Biogeochemical Processes, Max Planck Institute for Biogeochemistry, Jena, 07745, Germany.

[4]School of Integrative Plant Science, Cornell University, NY 14853, USA.

[5]State Key Laboratory of Earth Surface Processes and Resource Ecology, Faculty of Geographical Science, Beijing Normal University, Beijing 100875, China.

*Corresponding to*: Shuli Niu (sniu@igsnrr.ac.cn)

**Abstract:** Nitrogen (N) limitation greatly constrains terrestrial ecosystem carbon (C) uptake and its response to climate change and elevated carbon dioxide. Hence, accurate assessments of ecosystem N limitation are crucial for predicting C-N feedbacks, and vital for providing guidance for policy making or ecosystem management as well. This study aims to retrieve N limitation information by data model fusion from one field N addition experiment so that we can better understand N controls on the terrestrial C cycle. We estimated two sets of parameters with one C-only model and one coupled C-N model. Our results showed that the estimated leaf photosynthetic efficiency (LPE) and process rates (e.g., senescence and decomposition rates) of organic C from almost all pools were higher with the coupled C-N model than those with the C-only model at the ambient treatment. However, the differences in the LPE and the C exit rates between the coupled C-N model and the C-only model decreased with the increasing N addition rates. Both the C-only and coupled C-N models simulated similar C pool sizes as observed at every N addition treatment with their respective parameter estimates. However, simulated ecosystem C storage and gross primary productivity (GPP) decreased if we ran the coupled C-N model with the parameters estimated by the C-only model. This decrease was larger at the ambient treatment and became smaller with the increase of N addition. In general, we put forward a new method to retrieve N limitation information from observations by data model fusion. This method will make it possible to estimate the



global nutrient limitation and benefit ecosystem management and policy making.
**Keywords:** Data assimilation, nitrogen limitation information, nitrogen addition, model structure,
carbon and nitrogen cycles.
1.    **Introduction**
Nitrogen (N) availability is a key limiting factor for growth in many terrestrial ecosystems, and thus
important for both ecosystem productivity and the decay of dead organic material (Keeler et al., 2009;
Liu et al., 2019; Pregitzer et al., 2010). As a consequence, plant tissue N content is often highly correlated
with key metabolic rates such as photosynthesis (Zong et al., 2018) and respiration (Sun et al., 2014),
and an important control on the turnover of soil organic matter (Fog, 2008; Keeler et al., 2009).
Manipulative N addition experiments and field studies have demonstrated  larger plant growth with
increasing N deposition (LeBauer and Treseder, 2008; Pregitzer et al., 2010). Hence, the capacity of
terrestrial ecosystems to store carbon (C) is limited by its N availability and the C:N stoichiometry of
plant tissue (Hungate et al., 2003; Luo et al., 2004), especially under elevated atmospheric $CO_2$
concentrations.

An accurate assessment of ecosystem N limitation is crucial for predicting C cycle and its feedback

to climate change, which remain one of the biggest uncertainties in earth system models (Friedlingstein
et al., 2022). One important source of uncertainty in predicting C cycle is the degree to which N limits
plant growth (Elser et al., 2007; Hungate et al., 2003). Furthermore, the N limitations in plant growth,
photosynthetic capacity, and decomposition rates in litter and soil are poorly understood partially because
they are very difficult to be measured (Vicca et al., 2018). Some methods have been used to infer nutrient
limitation, including fertilization experiments, leaf nutrient resorption efficiency and the thresholds of
leaf N:P ratios (Bracken et al., 2015; Du et al., 2020; Sullivan et al., 2014; Tessier and Raynal, 2003).
But most of these methods are either very time-consuming and laborious, or have greater uncertainty,
although researchers have invested a lot of efforts to use these methods to retrieve nitrogen limitation
information in terrestrial ecosystems.

Data assimilation is a statistically rigorous method for estimating the parameter values of a mechanistic

model representing rates of transfer of C and N within an ecosystem. Not only does this method allow
models to be better calibrated to data, but it also provides great opportunities to understand model
parameterization that reflects N-limitation. When data assimilation is applied to calibrate models with





specific observations, the information in observations is integrated into the model via a set of specific
parameters (Luo and Schuur, 2020). All data assimilation studies indicate that the optimal estimated
parameters vary across different treatments of global change experiments (Liang et al., 2018; Luo and
Schuur, 2020; Wang et al., 2021; Xu et al., 2006). Because there are always some processes at unresolved
scales (processes that can't be represented explicitly) that may potentially interact with processes at
resolved scales (processes that can be represented explicitly) to influence model results. Varying
parameters is a useful modeling approach, which recognizes that the model need not explicitly
incorporate all processes on a resolved scale (Luo and Schuur, 2020). Meanwhile, the estimated
parameters are the result of retrieving the ecological information from the specific data set. Hence, many
studies used data assimilation to understand ecological processes (e.g., coefficients of plant allocation
and litter decomposition) by comparing the posterior probability density functions of parameter values
among different experimental treatments or sites (Liang et al., 2018; Wang et al., 2021; Xu et al., 2006).
Therefore, data assimilation provides the great possibility to retrieve N limitation information from
observations since the N limitation information is represented in parameterization as a model is calibrated
with the observations (Luo and Schuur, 2020). However, when a N cycle module is incorporated into the
C-only model, N processes are explicitly represented and simulated. N limitation influences on C
processes are no longer accounted by C-related parameter values (Wang et al., 2022). A study using data
assimilation technique found that parameter values change with model structures, while simulated
ecosystem C dynamics were similar (Wang et al., 2022), but they didn't further explore the nutrient
limitation information behind models with different structures and how it influenced the C cycle
predictions. Previous studies also tried to use models with or without a nutrient module to represent
whether there was the corresponding nutrient limitation (i.e., nitrogen and phosphorus) with data
assimilation (Du et al., 2021). But they ignored the fact that the nutrient limitation information in the C-
only model and C-nutrient coupling model is consistent because the observations they used for data
assimilation were identical. Hence, it's necessary to retrieve information on nutrient limitation by data
assimilation.
In this study, we used data assimilation to estimate parameters that reflect N limitation in a C-only
model and coupled C-N model from 17 data sets collected at a field N enrichment experiment at an alpine
meadow in the Qinghai-Tibet Plateau. The Bayesian probabilistic inversion as the data assimilation



method was used in this study to estimate the C cycle associated parameters and simulated C pool
dynamics of different ecosystem components under the integral measurement collected at Hong Yuan
alpine meadow field site from 2014 to 2020. The specific questions we addressed in this study are: (1)
how to retrieve N limitation information from experimental data by data assimilation? (2) how does N
limitation influence the predictions of ecosystem C dynamic?
**2.    Materials and methods**
**2.1 Site description**
The Hong Yuan station was located on the eastern Qinghai-Tibet Plateau (32°84'N, 102°58'E), which is
high elevation continental plateau with a frigid temperate monsoon climate. The mean annual
precipitation is 747 mm, the mean annual temperature is 1.5℃, the sunshine duration per year is about
2000-2400 hours, and the growing season lasts from April to October. The main vegetation type in this
study area is alpine meadow, and the soil type here is subalpine meadow and boggy soil (Song et al.,
2014). This area is dominated by *Deschampsia caespitosa* (Linn.) *Beauv., Koeleria cristata* (Linn.) Pers.,
*Gentiana sino-ornata* Balf. f., *Potentilla anserina* L., and *Anemone rivularis* Buch.-Ham (Quan et al.,

2018).

**2.2 Data source**
Data sets used to drive the Grassland ECOsystem (GECO)  model and used to estimate parameters in
this study both were from a N addition experiment and a co-located eddy-covariance measurement
system. Meteorological variables, such as soil volumetric water content (VWC) and soil temperature
(Tsoil) simultaneously measured with the eddy covariance system at a depth of 10 cm every hour.
Meanwhile, the photosynthetically active radiation, wind speed, relative humidity, and air temperature
used for simulating photosynthesis also come from the continuous observation of the eddy covariance
system. The N addition experiment near the Eddy covariance tower used a random block design with six
N addition treatments (N0, N2, N4, N8, N16, N32, representing N addition rates of 0, 2, 4, 8, 16, 32 g
$N \cdot m^{-2} \cdot year^{-1}$, respectively, with five replications each). The area of each plot is 8 × 8 m, and the distance
between adjacent quadrats is 3 m. During the growing season, from May to September every year, N was
added once a month. The method of N addition was spraying, and ammonium nitrate ($NH_4NO_3$, analytical
purity, content ≥ 99%) was used as N fertilizer. Gross primary productivity (GPP) and soil respiration



(SR) were measured twice a month with static chambers (LI-6400XT, LI-COR Environmental, Lincoln,
Nebraska, USA) in plots with different treatments in the growing season from 2014 to 2020. Biometric
measurements were made once a year to determine leaf and root biomass, standing and surface litter
quality, and microbes, soil C content, N content of leaves and roots, standing litter, surface litter, total N
content of microbe and soil, and soil inorganic N concentration in all plots.

The data that used to drive the model includes daily soil moisture, soil temperature, photosynthetically

active radiation, wind speed, relative humidity, and air temperature from 2014 to 2020. The data
assimilated into the GECO model for parameter estimation include C and N contents in leaf, root,
standing litter, surface litter, microbial, soil, and autotrophic and heterotrophic respiration at every N
addition treatment (Table S1).
**2.3 Model**
The GECO model was used in this study (Wang et al., 2021), which has evolved from the Terrestrial
ECOsystem (TECO) model (Xu et al. 2006, Shi et al. 2016) with a distinct standing litter pool for
grassland ecosystems. The GECO model has both coupled C-N and C-only version. There are seven C
and N pools and one more mineral nitrogen pool in the coupled C-N model. The pools are leaf (X1, N1),
roots (X2, N2), standing litter (X3, N3), surface litter (X4, N4), fast (X5, N5), slow (X6, N6), passive
soil organic matter (SOM, X7, N7) and mineral N pool (Fig. 1). But the C-only model just has the
counterpart C pools. In the GECO model, $CO_2$ in the atmosphere enters the ecosystem by canopy
photosynthesis which is simulated by a two-leaf model (Wang et al., 1998). The plant photosynthetic
capacity is limited by the leaf N concentration in the coupled C-N model, reflecting plant investment in
photosynthetic machinery for light harvesting and carboxylation rates (Leuning et al., 1995; Walker et
al., 2015). Leaf photosynthetic efficiency (LPE) limited by foliage N content as a scalar:
$$SNvcmax = \max\left(\min\left(\exp\left(-\frac{CN_{leaf}-CN_{leaf,0}}{CN_{leaf,0}}\right), 1\right), 0\right) \qquad (1)$$

where $CN_{leaf}$ is the estimated C:N ratio of foliage and $CN_{leaf,0}$ is the defined C:N ratio of foliage without
N limitation. Some of the photosynthates were used for plants' respiration, and the remaining assimilated
C was allocated to leaf (X1) and root (X2) pools. Detritus from the dead plants then flowed into the litter
pool, which contained standing litter (X3) and surface litter (X4). Some of the subsurface litter was
respired by microbes, while the rest was converted to fast SOM (X5) and slow SOM (X6). The $CO_2$



released by the decomposition of soil C eventually re-enters the atmosphere. Similarly, plants uptake N
from the mineral soil. Subsequently, the uptaken N is distributed to the plant pools and then transferred
to the litter and soil pools. The organic N in the seven pools returns to the soil through microbial
mineralization. The GECO model uses a matrix-based 1st-order differential equation approach to
describe the process of carbon transfer between ecosystem carbon pools as:
$$\frac{d}{dt}\boldsymbol{X}(t) = \boldsymbol{A}\xi(t)\boldsymbol{K}\boldsymbol{N_s}\boldsymbol{X}(t) + \boldsymbol{b}u(t) \tag{2}$$
$$\boldsymbol{X}(0) = \boldsymbol{X_0}$$
where $\boldsymbol{X} = (x_1\ x_2\ x_3\ x_4\ x_5\ x_6\ x_7)^T$, in which $x_{(t)}$ represents the C pools in leaves, roots, standing litter,
surface litter, fast, slow and passive SOM at time $t$. The matrix $\boldsymbol{A}$ represents C transfer between pools
(Xu et al., 2006). $\boldsymbol{K}$ is a 7×7 diagonal matrix with diagonal entries. The elements on the diagonal indicate
the C processing rates of each pool (i = 1, 2, ..., 7). $\boldsymbol{N_s}$ is a 7×7 diagonal matrix with diagonal entries,
elements on the diagonal indicate the N limiting effects on the pools decomposition rates, which is
represented by $\boldsymbol{N_s(i)} =$ exp((CN0-CN(i))/CN0) (i = 1, 2, ..., 7). For the C-only model, there is no any
diagonal matrix to represent the N limitation effects. $u$ represents the C produced by canopy
photosynthesis, which is constrained by equation (1). $\boldsymbol{b}$ is a vector of partitioning coefficients of
photosynthetic products to leaves and roots. $\xi(t)$ is an environmental scalar to account for effects of
temperature and humidity on decomposition (Luo et al., 2003).

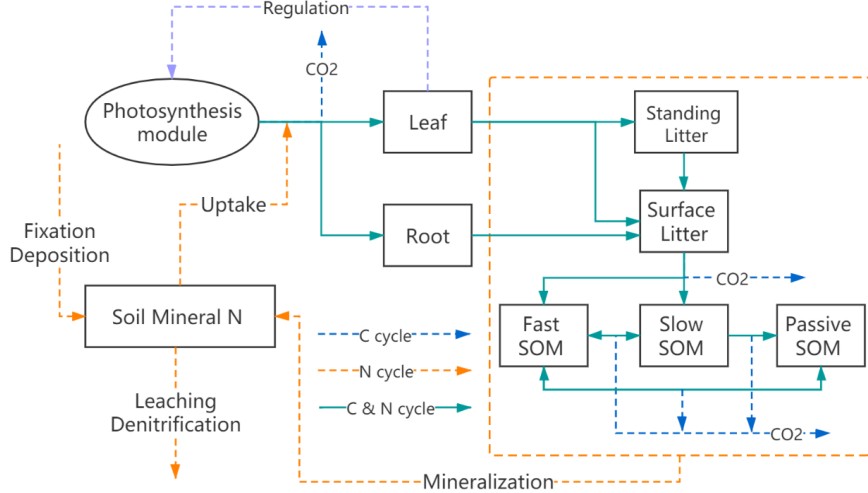

**Figure 1 Carbon and nitrogen pools and flux pathways in GECO model. Blue arrows show carbon transfer**
**processes, yellow arrows indicate nitrogen transfer processes, and green arrows represent C and N coupling**



**processes. SOM, soil organic matter.**
In the coupled C-N model ,the N processes can be described by this formula:
$$\frac{d}{dt}\boldsymbol{N}(t) = \boldsymbol{A}\xi(t)\boldsymbol{KN_s R^{-1}X}(t) + \kappa_\mu N_{min}(t)\boldsymbol{\pi} \tag{3}$$

$$\mathbf{N}(0) = \boldsymbol{N}_0$$

where $\boldsymbol{N}$ = (n$_1$ n$_2$ n$_3$ n$_4$ n$_5$ n$_6$ n$_7$) $^T$, in which n$_{(t)}$ represents the N pools in leaves, roots, standing litter,
surface litter, fast, slow and passive SOM at time $t$. $\boldsymbol{R}$ is a 7×7 diagonal matrix with diagonal elements
indicating the C:N ratio of each pool. $\boldsymbol{\pi}$ = ($\pi_1$ 1-$\pi_1$ 0 0 0 0 0)$^T$ is an allocation coefficient vector of N from
mineral soil to leaves and roots. $\kappa_\mu$ is N uptake rate of plants, $N$ min($t$) is the amount of available N in
the soil at time $t$. The dynamic equilibrium of mineral N in soil is determined by the input of
mineralization, biological fixation, atmospheric deposition, the output of plant input, leaching, and
gaseous N fluxes, which can be described by:
$$\frac{d}{dt}N_{min}(t) = -(\kappa_u + \kappa_L)N_{min}(t) + \boldsymbol{A}\xi(t)\boldsymbol{\varphi_1^*}\boldsymbol{KN_s R^{-1}X}(t) + F(t) \tag{4}$$

$$N_{min}(0) = N_{min,0}$$

In formula (4), $\kappa_\mu$ and $\kappa_L$ represent rates of N uptake and N loss, respectively. $\boldsymbol{A}\xi(t)\boldsymbol{\varphi_1^*}\boldsymbol{KN_s R^{-1}X}(t)$
represents N mineralization, $\boldsymbol{\varphi_1^*}$ represents mineralization rate, and $F(t)$ represents N input by biological
fixation and atmospheric deposition.
In this study, the initial pool sizes of leaves, roots, standing litter, surface litter, fast soil, slow soil, and
passive soil pools were constrained using ambient treatment data. The same values used in the ambient
treatment were used for initial pool sizes in the N addition treatment, assuming that there was no
significant difference between the ambient and supplemental N treatments before treatment began.
Compared with the coupled C-N model, the C-only model doesn't contain any N-cycle process. In
this study, the C exit rates of 7 pools, the C allocation coefficients of GPP, and the C transfer coefficients
were estimated using both the C-only model and the coupled C-N model. Meanwhile, N-related
parameters such as N partitioning coefficient, N uptake, N loss, external N input, initial mineral N pool,
and C:N ratios of different ecosystem components were estimated in the coupled C-N model.
**2.4 Data assimilation**
We used Markov-Chain Monte-Carlo (MCMC) to estimate parameters values of the GECO model. In
this method, the targeted parameters are considered as random variables within to a certain prior
probability distribution. According to the Bayesian theorem, the prior knowledge about the parameters





and the information contained in the data are fused to generate posterior distributions of the parameters
(Xu et al., 2006) as

$P(p|Z) \propto P(Z|p)P(p)$                                        (5)

In formula 5, $P(p)$ and $P(p|Z)$ represent the prior probability density function (PDF) and posterior
probability density function (PPDF) of parameters, respectively. $P(Z|p)$ represents conditional
probability density of observation under the prior parameters, which is also called the likelihood function
of $p$. We assume that the random error is normally distributed and has a mean of zero, so the likelihood
function can be represented as follows:

$P(Z|p) \propto \exp\left\{-\sum_{i=1}^{17}\sum_{t\in Z_i}\frac{[Z_i(t)-\varphi_i X(t)]^2}{2\sigma_i^2(t)}\right\}$                      (6)

In formula 6, $Z_i(t)$ and $\varphi_i X(t)$ represent the measured value and simulated values of the observed
variable $i$ at time $t$, and $\sigma_i$ is the standard deviation of the observed variable $i$. In this study, $i$ from 1 to
17 represents seventeen data sets, which are the C or N contents of leaves, roots, standing litter, surface
litter, microbes, mineral soil and heterotrophic respiration, inorganic N in soil, soil mineralization, N
uptake by plant and external N input. $\varphi_i$ is the mapping vector that maps the simulated state variables to
the observed data. For example, the observation operator $\varphi$ is expressed as follows:
Leaf C and N: $\varphi_1 = (1\ 0\ 0\ 0\ 0\ 0\ 0)$
Root C and N: $\varphi_2 = (0\ 1\ 0\ 0\ 0\ 0\ 0)$
Standing litter C and N: $\varphi_3 = (0\ 0\ 1\ 0\ 0\ 0\ 0)$
Surface litter C and N: $\varphi_4 = (0\ 0\ 0\ 0.5\ 0\ 0\ 0)$
Microbial C and N: $\varphi_5 = (0\ 0\ 0\ 0\ 1\ 0\ 0)$
Mineral soil C and N: $\varphi_6 = (0\ 0\ 0\ 0\ 1\ 1\ 1)$

The Metropolis-Hastings (M-H) algorithm was used as the (Hastings, 1970; Metropolis et al., 1953)

MCMC sampler. The initial parameter set was randomly selected within the priori parameter ranges. At
each iteration, a set of parameters ($p_{new}$) is proposed based on the accepted parameters from the previous
iteration ($p_{k-1}$). We accept $p_{new}$ only if $R = \frac{P(p^{new}|Z)}{P(p^{k-1}|Z)} >$ a random number between 0 and 1.
Otherwise, $p_{new}$ will be rejected, and we let $p_k = p_{k-1}$ to start the sampling of next iteration. The M-H
algorithm is repeated until 300,000 sets of parameter values have been accepted, and then all accepted
parameter values are used to construct the probability distribution functions (PDFs) (Weng and Luo,
2011; Xu et al., 2006).
**3.  Results**

## 3.1 Model performance on simulating C cycle under different N addition gradients

We selected 100 sets of parameter values from the PPDFs to run the GECO model and simulate the C dynamics from 2014 to 2020 using the C-only model and the coupled C-N model, respectively. The two models simulated the C pools in leaves, standing litter, surface litter, microbes, soil, and GPP well compared to observations under the different N addition treatments (Figure 2 and Figure S1-S2).

Meanwhile, among 19 C-related parameters in both the C-only and coupled C-N models, nine were well constrained by observations according to their posterior PDFs. These nine well-constrained parameters are baseline leaf maximum carboxylation rate, C exit rates of root, leaf, standing litter, surface litter, fast SOM, slow SOM, the allocation coefficients of C to root and leaf under all treatments. While the C exit rate of the passive SOM and the transfer coefficients among pools ($f_{i,j}$) were poorly constrained. In this study, we used the comparison of well estimated parameters between two models to indicate the N limitation in the C cycle processes represented by these parameters, and used the comparison of different C pool sizes between different models to indicate N limitation in the C pools as well.

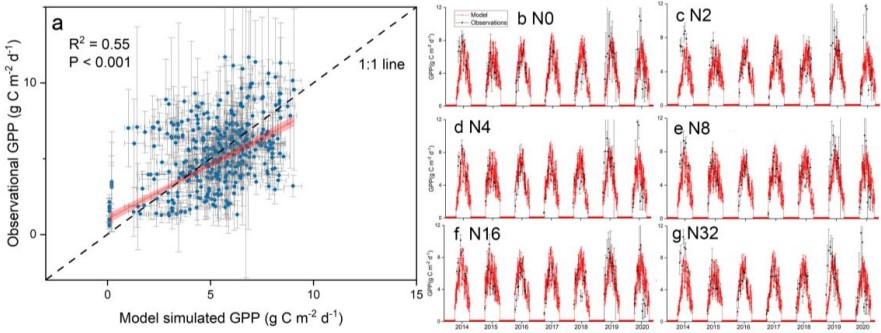

**Figure 2 Comparison of observed and model-simulated GPP simulated by the coupled C-N model. A is the overall comparison under different addition rate; b, c, d, e, f, g are the time dynamics of observed (dots) and model-simulated (red lines) GPP simulated by the coupled C-N model under N0, N2, N4, N8, N16, N32 addition rate, respectively.**

## 3.2 Estimated parameters between model structures and N addition treatments

The C-only and coupled C-N models led to different posterior PDFs of these well-constrained parameters under the different treatments (Figures 3 and 4). The leaf photosynthetic efficiency (LPE) was higher in the coupled C-N model than in the C-only model. But changes in estimated parameters between the C-only and coupled C-N models differed with different N addition treatments. The estimated LPE decreased with increasing N addition in the coupled C-N model, but the estimated LPE increased first



and then decreased with the increase of N addition in the C-only model. The divergent responses to N
addition treatments led to the results that the differences in the LPE between the coupled C-N model and
the C-only model got smaller with the increase of N addition rates (Figure 3).
C exit rates of all the pools were higher in the coupled C-N model than in the C-only model at the
ambient treatment (Figure 4). And with the increase of N addition rate (from N0 to N8), the differences
in the C exit rates of all the pools except the surface litter and fast SOM pools between the coupled C-N
model and the C-only model got smaller. But with the continuous increase of nitrogen addition (N16 &
N32), the differences of parameters between the C-only model and the coupled C-N model no longer
decreased, or even became greater. In general, the N addition effects on these parameters were consistent
in the C-only and coupled C-N model. N addition effects on parameterization varied among parameters.
C exit rate of the root pool decreased with the N addition rates, whereas the C exit rates of the standing
litter, fast SOM and slow SOM pools decreased first and then increased with the N addition rates. For
the remaining parameters, N addition didn't significantly change them (Figure 4).

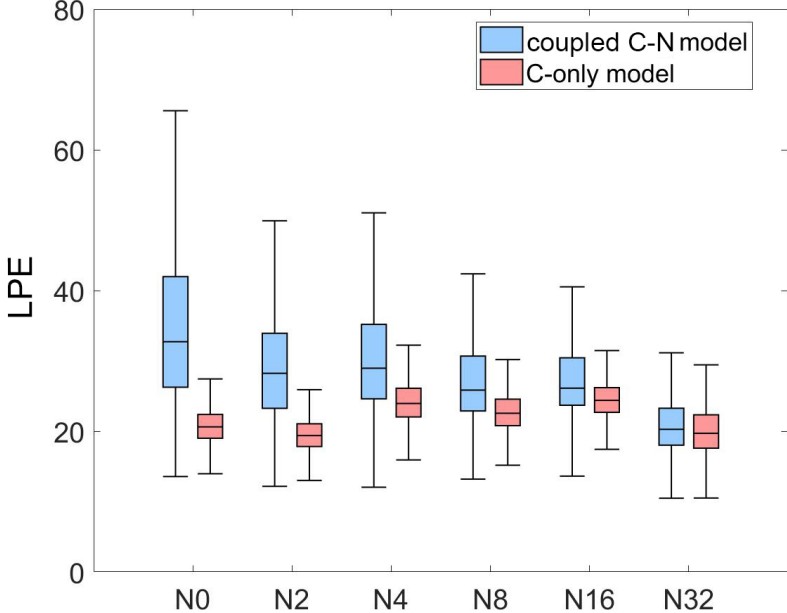


**Figure 3 Posterior distributions of estimated leaf photosynthetic efficiency (LPE) of the C-only and coupled**
**C-N models under N0, N2, N4, N8, N16, N32 addition rate, respectively.**



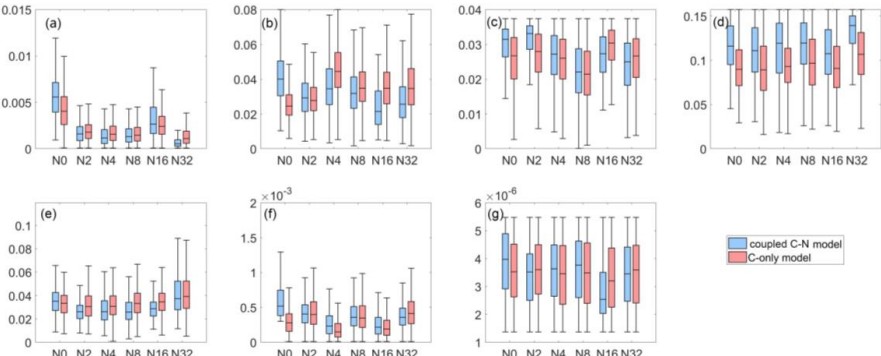

**Figure 4 Posterior distributions of estimated key parameters of the C-only and coupled C-N models under different N addition treatments. Baseline senescence rates of fine root (a) and leaf (b); baseline decomposition rates of standing litter (c), surface litter (d), fast SOM (e), slow SOM (f), and passive SOM (g). The red and blue boxes represent the distributions of estimated parameters of the C-only and coupled C-N models, respectively.**

**3.3 Simulations of C dynamics with the C-only and coupled C-N models**

The different ecosystem C pools exhibited divergent responses to N addition treatments in the simulations by both the C-only and coupled C-N models. Simulated passive SOM pools showed no N addition effects (Figure 5g). Except for passive SOM pool, most of the rest carbon pool size increased first and then decreased with the increase of N addition (Figure 5). Simulated leaf and surface litter C pools reached the maximum pool size at N16 treatment (Figures 5a, 5d). Simulated standing litter reached the maximum pool size at N8 treatment (Figure 5c). However, simulated fast and slow SOM C pools reached the maximum pool size at N4 treatment (Figures 5e, 5f). Because the slow SOM C accounts for about 90% of the total soil C and accounts for about 70% of the total ecosystem C, total soil C and ecosystem pools responded similarly with slow SOM under the N addition treatments (Figure 5h, 5i). Meanwhile, simulated GPP also showed a unimodal response with the increase of N addition and the maximum GPP values appeared at N16 treatment in both the C-only and coupled C-N model (Figure 6).

To further elucidate the N effects in the C only model, we tested the response of the coupled C-N model without retuning parameters to compare the N limitation of ecosystem carbon pools and flux due to the addition of N coupling components. The results showed that the simulated ecosystem C storage and GPP decreased if we ran the coupled C-N model with the parameters estimated by the C-only model (Figures 5 and 6). In addition, the decreased simulations by the coupled C-N model with the parameters estimated by the C-only model were greater at the ambient treatment, and the decreases were reduced



with the increase of N addition (Figures 5 and 6).  However, when the N addition reached at the N8
treatment, the difference between simulations by the coupled C-N model with the parameters estimated
by the C-only model and simulations by the coupled C-N model with its count part parameters became
very small. Hence, the simulations of the C-only, coupled C-N models and the coupled C-N model with
the parameters estimated by the C-only model were consistent at high N addition treatments
(Figures 5 and 6).

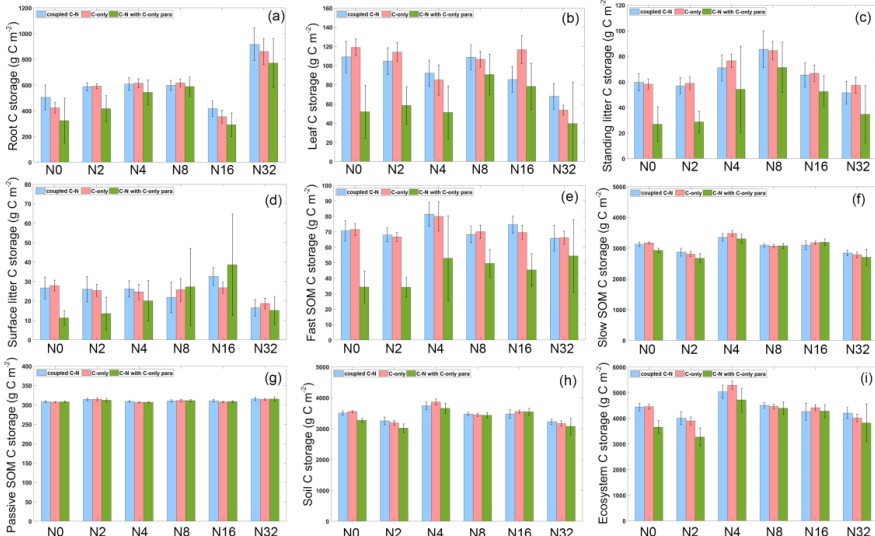


**Figure 5 Model simulated ecosystem C storage under different N addition treatments in 2020. They are root**
**(a), leaf (b), standing litter (c), surface litter (d), fast SOM (e), slow SOM (f), passive SOM (g), total soil C**
**storage (g) and total ecosystem (i) simulated by the C-only model with the C only parameters (blue bars), the**
**coupled C-N model with the coupled C-N parameters (red bars) and the coupled C-N model with the C only**
**parameters (green bars) under different N addition rates**





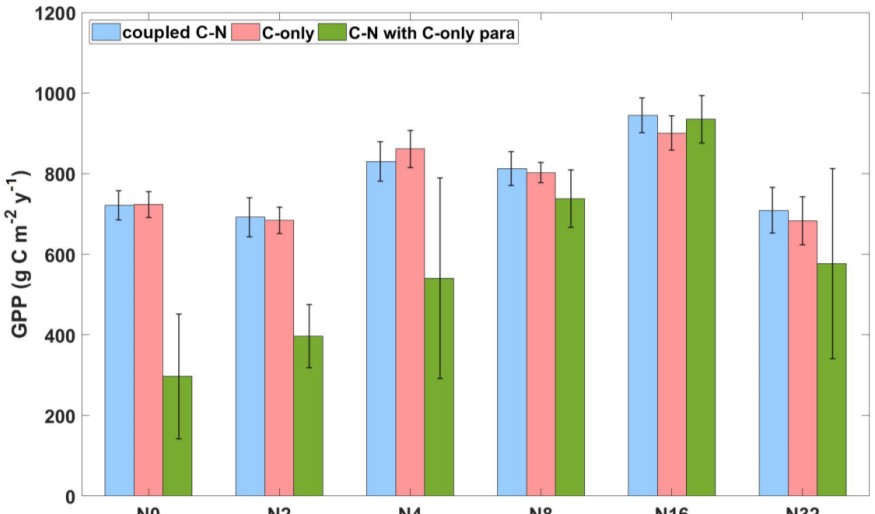

**Figure 6 Model simulated GPP under different N addition treatments in 2020. Blue bar is ecosystem C storage simulated by the C-only model with C-only parameters; Red bar is ecosystem C storage dynamic simulated by the coupled C-N model with coupled C-N parameters; Green bar is ecosystem C storage dynamic simulated by the coupled C-N model with C-only parameters.**

## 4. Discussion

### 4.1 Retrieving N limitation information by data assimilation and its implications

In this study, we used data assimilation to retrieve N limitation information from observations. Our results indicated that N limitation information can be retrieved by the following ways. First, the differences between the parameters estimated by the C-only model and the coupled C-N model with the same observation set. The parameters estimated by models with the data assimilation technique can represent the C cycle processes of an ecosystem, e.g., GPP allocation to leaf and root are represented by two allocation coefficients, and soil decomposition is represented by the soil exit rate. Because in the C-only model, the N-related information contained in the observation is implicitly represented in the estimated parameters. But when a coupled C-N model is calibrated using the same observations, the N-related information is no longer implicitly represented in the C-related parameters (Wang et al., 2022). Second, the differences between the C pool dynamic simulated by the coupled C-N model with parameters from C-only model and the C pool dynamic simulated by the coupled C-N model with its counterpart parameters (Figure 7), because the N limitation information behind parameters estimated by the C-only model can be intuitively expressed by the simulations of ecosystem C pools. And our method in

quantifying N limitation degree was confirmed by the N addition gradient in this study.

We highlight two main advantages of our method to retrieve nutrient limitation. On the one hand, by

comparing the parameters shift and pools size, we can evaluate the N limitation effect on any specific C
cycle processes of an ecosystem. And the N limitation degree can be quantitatively reflected by the
differences between estimated parameters or simulated pool size. On the other hand, the data set used for
data assimilation is no longer limited to nitrogen addition experiments. All of the observations which
contain the basic C and N condition can be used to evaluate the N limitation by a C-only model and the
coupled C-N model with data assimilation technique. In addition, there have been studies doing inverse
analysis using global data products (Bloom et al., 2016; Yang et al., 2021), it will become possible to
estimate the global nutrient limitation distribution by our method with increasing global data products
with high reliability.

Due to the important effect of N on ecosystem C dynamics, it is vital to evaluate N limitation degree

of an ecosystem. Understanding the N limitation condition at different areas can provide guidance for
policy making and ecosystem management. Different managements should be conducted according to
their N restriction condition of a specific ecosystem, so as to manage an ecosystem most efficiently.
Besides, the N limitation in different C cycle processes and pools were poorly understood because it is
hard to measure (Vicca et al., 2018). Our method can retrieve N limitation information of any specific
ecosystem C cycle processes or pools, by which we can adjust ecosystem functions and services more
accurately by precise management.

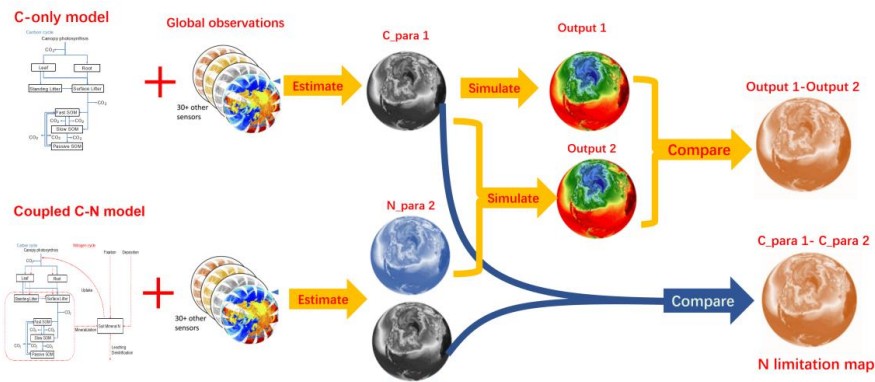

**Figure 7. A concept figure about how to map the global N limitation of a specific C pool.**
**4.2  Retrieve nitrogen limitation  information from different C cycle processes**



In this study, we used two models: The C-only and coupled C-N models, to retrieve information on N
limitation. Our results show that the senescence rate from leaf and root, the decomposition rates of
standing litter, surface litter, fast SOM, slow SOM and passive SOM, the allocation coefficients of GPP
to leaf, and the LPE were higher in the C-N model than these in the C-only model at the ambient treatment
and lower in the N addition treatments (Figures 3, 4 and Figure S3). The N limitation degree was the
strongest at the ambient treatment, but the N regulations of C cycle processes are not explicitly simulated
and, thus, considered to be unresolved processes in the C-only model (Wang et al., 2022). Hence, the N
limitation on photosynthesis was reflected in a smaller value of LPE in the C-only model than that in the
coupled C-N model which incorporated a N module into the C model. Because N processes were
explicitly simulated in the coupled C-N model, N limitation effects on C processes were no longer
accounted by C-related parameter values (Figure 3). Meanwhile, the N limitation degree at the high N
addition treatments was much less than that at the ambient treatment (Figure S4). Therefore, differences
in the LPE between the coupled C-N model and the C-only model got smaller with the increase of N
addition rates (Figure 3), indicating that N limitation degree was reduced or even disappeared with the
increase of N addition rates.  It needed to be specifically pointed out that we didn't consider any toxic
effect at the high N addition treatment because there was usually less N deposition in the real world than
in most field experiments (Adams et al., 2021).
Similarly, the senescence rates from leaf and root, the decomposition rates of standing litter, surface
litter, fast SOM, slow SOM and passive SOM were regulated by their counterpart C:N ratios (Niu et al.,
2010), and this regulation was also reflected in this model (Wang et al., 2022). The N limitation on their
senescence rates and decomposition rates were reflected in smaller values in the C-only model than that
in the coupled C-N model. Different from LPE, N limitation degree in those variables  between the
coupled C-N model and the C-only model got similar at the middle N addition rates (N2, N4, N8).
Because low-level N additions could alleviate N limitation but high-level N addition would harm plants
due to the ionic toxicity (Aber et al., 1998; LeBauer and Treseder, 2008; Niu et al., 2016). That was why
the differences in some parameters estimated by the coupled C-N model and the C-only model even got
larger at high N addition treatments. However, some other parameters estimated by different models had
no significant difference in different N addition treatments. This may be due to the fact that these
parameters contain little N limitation information, and therefore these C cycle related processes are not





regulated by N processes and can be considered to be resolved processes in both the C-only and coupled
C-N models.

**4.3  Retrieve nitrogen limitation  information from different ecosystem C pools**

Although the N limitation information behind many estimated parameters was different, two models
simulated similar pool sizes when two sets of parameters were used in accordance with their different
structures (Figure 5). This was reasonable as models with different structures were supposed to similarly
simulate the dynamics of the same ecosystem under the same conditions (at every N addition treatment).
When we incorporated an N module into the C model, N processes are explicitly simulated, estimated
C-related parameters no longer contain N processes for the coupled C-N model, and the N limitation
information was retrieved from the C-only model as well.

By using the coupled C-N model with parameters estimated by the C-only model to simulate the

terrestrial ecosystem C dynamic, the N limitation information behind parameters can be intuitively
expressed. Our results showed that GPP, ecosystem C storage in plant and soil pools simulated by the
coupled C-N model with the parameters estimated by the C-only model all decreased at the N0 treatment
(Figures 5 and 6), but these C pools and GPP change got smaller with the increase of N addition (Figures
5 and 6). These results indicated that the parameters estimated in the C-only model did contain the N
limitation information, and the limitation degree reduced with increasing N addition rates. If there was
no N limitation in observations, the parameters estimated by either the C-only model or coupled C-N
model would not contain any N limitation information. Therefore, the C dynamic simulated by the
coupled C-N model with the parameters estimated by the C-only model would be consistent to the result
simulated by the C-only model. If there was N limitation in observations, the parameters estimated by
the C-only model would contain the N limitation information but the parameters estimated by the coupled
C-N model would not. Our results that the C fluxes and pools simulated by the coupled C-N model with
the parameters estimated by the C-only model was smaller than that simulated by the C-only model,
indicate clear N limitation information. The stronger the N limitation, the greater the decrease of the
simulation by the coupled C-N model with the parameters estimated by the C-only model (Figures 5 and

6).

Given that parameter values in their original C-only models were kept or manually tuned for the

new models with N processes (Koven et al., 2013; Sokolov et al., 2008; Zaehle and Friend, 2010),



predictions by the newly modified ESMs with the N modules mostly predict lower photosynthesis rates,
lower C sequestration, and lower ecosystem C storage than their C-only counterpart models. Our results
in this study reveal that such predictions of lower C storage with the coupled C-N models than their
original models may not reflect reality (Figure 8). In contrast, ESMs may underestimate both future
terrestrial C sequestration and the potential C-climate feedback because they overestimate N constraints.

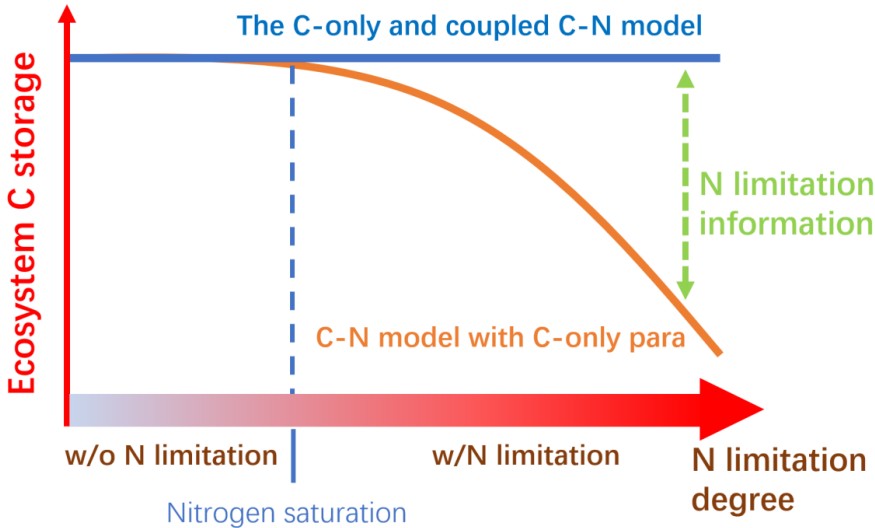


**Figure 8 The influence of different N limitation degree on the parameterization of the C-only model and the**
**coupled C-N model. Blue line is the C storage simulated by the C-only model or coupled C-N model with data**
**assimilation, orange line is the C storage simulated by the coupled C-N model with parameters estimated by**
**the C-only model.**
5. **Conclusion**
Based on a 7-year field N addition experiment and data assimilation method, this study carried out an
inverse analysis with the C-only model and the coupled C-N model. We found that data assimilation
technique could be used to retrieve N limitation information from observations. N limitation information
of some C cycle processes can be retrieved by comparing the differences between the parameters
estimated by the C-only model and the coupled C-N model with the same observation. N limitation
information of the C pools and fluxes can be retrieved by comparing the differences between the C pool
dynamic simulated by the coupled C-N model with C-only parameters and the C pool dynamic simulated
by the coupled C-N model with its counterpart parameters. In addition, N additions had a unimodal
response on most ecosystem component C storage which was mainly determined by their exit rates in



this study. In general, we put forward a new method to retrieve N limitation information from
observations by data assimilation technique. With the increase of global data products, our method will
make it possible to estimate ecosystem nutrient limitation with high reliability and thus provide guidance
for policy making or ecosystem management.

**Data availability**
All additional data produced in this study will be published FigShare (10.6084/m9.figshare.22094111)

**Author contribution:** SW, JW and SN designed the experiments, evaluated the data and wrote the
manuscript. CS, YL were involved in the writing (review and editing) of the paper before submissions.
WC collected and arranged data. YZ, AY developed the conceptualization and methodology of this study.

**Competing interests:** The authors declare that they have no conflict of interests.
**Acknowledgments**
This study was supported by National Key R & D Program of China (2022YFF0802102) and National
Science Foundation of China (31988102)

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
