# Peer review of "Nitrogen limitation information retrieved from data assimilation"

_Biogeosciences, 2023_

## Author Comment (AC1)

**Responses to comments of Reviewer #1:**

We thank Referee #1 for insightful and constructive comments that help improve the manuscript. We have carefully considered the constructive comments and suggestions from the reviewers, and we provide answers to them here accordingly. Reviewer's comments are in black, our replies are in blue.

**General Comments:**

The paper "Nitrogen limitation information retrieved from data assimilation" by Wang et al. proposes a method to apply data from nitrogen addition experiments in combination with calibration of C-only and CN-coupled models to obtain information on nitrogen limitation. By comparing the calibrated values of either model setup and by comparing the predicted carbon pools, they claim to identify ecosystem processes that are responsible for nitrogen limitation in any given vegetation model. The paper tries to disentangle different carbon-nitrogen feedbacks to guide future model development regarding potential nitrogen limitation under elevated CO2.

However, the paper shows three major issues: (1) The methodology and results are not suitable for inferring main conclusions. (2) The paper has multiple insufficiencies regarding the quality of science and presentation (e.g., unclear use of concepts and missing definitions, imprecise wording, and hard-to-read display items). (3) The use of references is inadequate (e.g., methodology and Figure 1 are largely copied from a published but not referenced paper, referenced publications are mostly from authors, lack of references to remaining literature, and some references in the main text are not in the bibliography).

Moreover, most of the criteria of Biogeosciences are not fulfilled. Even if these major issues are addressed, the paper is unlikely to be worthy of publication. Therefore, this paper is recommended to be rejected.

Response: We gratefully appreciate your valuable comments, which are very helpful for us to improve the manuscript. Here is the summary for our responses to the three major comments. (1) We apologize for the unclear presentation of the methodology. One important assumption of our study is that we can learn about carbon cycle processes through model-data fusion. The technique itself is not new and it has been used in many other previous studies to learn about ecosystem processes using information from a model and experimental data (e.g. Shi *et al.* (2016), Guo *et al.* (2020) and Liang *et al.* (2018)). What is new in our study is the application of data assimilation to a nitrogen manipulation experiment, from which we aim to obtain information on the processes by which N limits primary production. Our approach consists on obtaining first parameters of the C cycle only by data assimilation. In this C-only model, C cycling processes (parameters) that are affected by ecosystem nitrogen availability are regarded as unresolved processes. In a second step, we used the C-N coupled model to estimate the values of these previously unresolved parameters, using information from the nitrogen addition experiment to learn about the processes that regulate the C cycle. By comparing the difference of these estimated parameters, we aim to quantify the degree by which nitrogen limitation contributes to these C cycle processes. We will improve the text to make the argument more clear in a revised version.

(2) We are really sorry for the unclear concepts, definition, imprecise wording, and

unclear display items. We will carefully check our manuscript and thoroughly improve the descriptions to make them clearer.
(3) We also realized that our citations are very limited. When revising the manuscript, we will pay more attention to this issue and will cite relevant previous studies and related literature.

**Major Issues**

- The paper lacks a sufficient description of its methodology and requires more explicit discussion and contextualisation of the results with respect to recent publications on nitrogen limitation processes, such as Zaehle et al. 2014, Jiang et al. 2019, Arora et al. 2020, Davies-Barnard et al. 2020. The text should be improved at the following positions:

L94: For readers not familiar with the GECO model, it would be useful if its processes were explained in more detail. More importantly, to derive any information on nitrogen limitation, the model's carbon-nitrogen feedback must be explained clearly (e.g., response to N-deposition, changes in carbon-to-nitrogen ratios of plant tissues, changes in allocation, nitrogen uptake). Also, without a process-based explanation of GECO's carbon-nitrogen dynamics and without comparison to representations in other Earth System Models, one cannot draw conclusions on whether current Earth System Models simulate N limitation realistically or not.

Response: Thank you for this valuable suggestion, we will add a more complete description of the GECO model, particularly those processes related to the interaction between the N and C cycles in the model. Also, to focus on the main results and the main argument we want to convey, we will delete the text about applying our method to Earth System Models.

L157: The definition of the N-limitation matrix is presumably smaller than 1 (not clearly defined in the methodology). This implies that the calibrated parameters of the CN-model will be larger than the parameters of the C-only model. This behaviour is visible in the results, and therefore the discussed difference in model parameters is a methodological consequence and holds no information on N-limitation. A strong argument needs to be provided why calibrated parameters from models with different structures should be directly comparable.

Response: We partly agree that our results about estimating parameters between models with different structures may lead to results prescribed by the model structure itself. However, we contend that the estimated parameters from the C-only model actually contain information on N-limitation. Because these parameters were estimated from observations from a N manipulation experiment, the parameters in the C-only model do contain information about how N-limitation affect the C cycle. By adding a N module to the C-only model and re-estimating parameters in the newly developed C-N coupled model, the N limitation information is transformed from average parameter values to a specific mathematical function. We focus on two main processes in our analysis, the effect of foliage N content on photosynthesis, and the N-dependent decay rate on decomposition, which are commonly included in carbon-nitrogen coupled models (Gerber et al., 2010; Zaehle and Friend, 2010). Following this comment, we will add more discussion to state this argument in the revised version of the manuscript.

L325: The proposed upscaling approach is not convincing because it lacks a more detailed explanation. The method seems not to be generalizable across space, which weakens any inference on global nitrogen limitation effects (as done in the abstract and conclusion). The

authors should clarify the following questions: What gives the confidence that other data assimilation studies do not need N addition data? How is a "nitrogen limitation degree" quantifiable? What are "basic C and N conditions" and "global data products", and how can the latter hold information on the former? How can results from different models be inter-comparable to achieve "global nutrient limitation distributions"?

Response: We agree with your comments about the application of our method of global scale analysis of global nutrient limitation. We also think more studies will be required before we upscale our method to a global scale. In the new version, we put this method in an ecosystem-scale context, and remove statements about a global-scale application.

As to the first two questions, one of the basic assumptions of this paper is that we can understand the status of the carbon cycle of the ecosystem through model-data fusion. In this case, we can use as many observations as possible to better understand ecosystems. At the same time, there are some carbon cycle processes that are not easy to observe, which can also be calculated by data assimilation, such as carbon allocation and transformation between pools, carbon turnover in specific pools, or decomposition rates. In this article, we analyzed nitrogen limitation by comparing the parameters retrieved from different structural models. For example, assuming that in an area with very limited nitrogen, the carbon content of the leaves is very low, in a carbon-only model, we would obtain a very low parameter related to photosynthetic rate to match the observed values. However, in a carbon-nitrogen coupled model, the same parameters may be constrained by an equation related to leaf nitrogen to match the observed values. In this case, the difference between the two parameters can reflect the degree of nitrogen limitation of the carbon cycle process.

"basic C and N conditions" is some observational data related to carbon and nitrogen cycles, like GPP, biomass, soil C and N content, etc. Based on these data sets, we can do data assimilation to understand C and N cycles at a larger scale, as Yang et al. (2021) and Bloom et al. (2016) did. To avoid confusion, we will change this description to "background C and N conditions.

For the comparison, the method introduced in this study can be used for paired models which contain a C-only model and a C-N coupled model developed from the C-only model. Both of them have the C-related parameters, and these paired parameters are comparable because they represent the same C-cycle processes.

L338: It is not apparent how the results can inform ecosystem services and management. This requires a more detailed explanation.

Response: Here we want to illustrate that the distribution of nitrogen limitation in space can provide a reference for ecosystem management. For example, if we try to transform an area into farmland, areas with unrestricted nutrition will be a better choice. And the nitrogen limitation map can provide useful information to ESMs, which may help the region or global carbon estimation. But to avoid confusion, we will delete this description.

L369: The authors state that some parameters are significantly different across different N addition treatments, whilst other parameters are not. This requires more detail and should address the following questions: Which are the exact parameters that are referred to here? Is it reasonable that the respective parameters are (not) influenced by nitrogen limitation? Does this agree with findings from other studies?

Response: Thanks for your suggestion, and we're sorry for the unclear description. The

parameters we mentioned here are standing litter, surface litter, and passive SOM that were not affected by model structure. For example, the decomposition rate of passive SOM didn't change in the N addition treatments, possibly due to the long response time of this pool to changes in N; therefore, the nitrogen limitation effect on this pool cannot be reflected at a short time scale. We will add more sentences here to discuss which parameters didn't change with N addition and why it happened.

L404: The final sentence of the paragraph makes a strong claim that current ESMs overestimate N constraints. This requires more argumentation than stating it is a consequence of not re-calibrating CN-models. What are the parameters that need to be re-calibrated most urgently? Are other studies also hinting towards too stringent N limitation in current ESMs?

Response: Thanks for the critical comments. We realized that we didn't use an earth system model in this study although most C and N processes and their coupling relationship were based on similar descriptions from Earth System Models. Hence, the claim that current ESMs overestimate N constraints is difficult to illustrate in this study. We decide to remove claims about the N limitation in ESMs in the revised manuscript.

Quality of presentation (Figure listed here, specific sections in text are listed in under "Minor Issues" below):

Figure 1: It is unclear what the light-blue "Regulation" arrow denotes.

Response: We will explain it in the "regulation" in the method part.

Figure 2b-g: Plots are generally too small. Axes are not legible. It is difficult to judge whether the predicted interannual cycle matched observations.

Response: Agreed, we will make it clearer when revising our manuscript.

Figures 4 and 5: Increase font size, respectively, size of the entire figure.

Response: Agreed. We will modify them.

Figure 7: Description of upscaling procedure should be more exhaustive because it is not evident from the concept figure alone.

Response: Thanks for your suggestion, we realized that we need more work to upscale our method to a global scale. In the new version, we put this method in an ecosystem-scale context, and removed statements about a global-scale application.

Regarding references

Intransparent re-use of published methods: Large parts (e.g. Figure 1) of the methodology are identical to Wang et al. 2022, which requires clear referencing.

Response: We apologize for the mistakes. we will add the references in the accordingly.

The bibliography does not include all references from the main text (e.g. Shi et al. 2016, Koven et al., 2013; Sokolov et al., 2008; Zaehle and Friend, 2010)

Response: We will add these references and others from relevant related studies.

Friedlingstein et al. 2022 (Global Carbon Budget 2022) is a carbon accounting paper and is not a suitable reference for the uncertain feedback of nitrogen limitation to climate change across Earth System Models. For example, Chapter 5 of IPCC (2021) and the references therein are more suitable.

Response: Agreed. We will change it.

The bibliography relies strongly on references from co-authors. A broader literature base is necessary to reflect critically on the results and to support conclusions.

Response: Thanks for your suggestion, we used references from co-authors mainly due to the fact that methods and ideas came mostly from these previous studies. But we value your suggestion and use a broader literature to discuss our results.

**Minor Issues**

- The paper often mixes different types of statements that should be separated:

L186: At the end of the model description, a key part of the method is described. This should appear towards the beginning of the section so that the reader knows why the model description is needed at all.

Response: Agreed, we will follow your suggestion.

L282: Here, a new methodology is introduced. This belongs to the methods and not to the results.

Response: Agreed, we will move the content to the methods part.

L331: This is an exact repetition of the relevance statement from the introduction. This should either be removed or shortened and moved to the beginning of the discussion as a short reminder for the reader.

Response: Agreed, we will reword these sentences.

The discussion holds multiple lines at which the same information (the finding that information on N-limitation is extractable from comparing calibrated parameters) is repeated. This should be largely condensed. Instead, the discussion should be used to discuss the plausibility of the findings and whether they support the conclusion or not. This crucial scientific process is missing.

Response: Thanks for your comments. We will condense the discussion on N limitation parameters and discuss more about the plausibility of the findings in a broader context.

The following lines require clarifications:

L51: What are "thresholds of leaf N:P ratios"?

Response: Thresholds of leaf N:P ratios is the relationship between plant N and P content, which is regarded to be an indicator of nutrient limitation (Du et al., 2020).

L61: It is unlikely that "all data assimilation studies" that exist were considered. A different adjective should be used, or a specific subset of data assimilation studies should be clarified.

Response: Agreed, we will reword this sentence.

L65:What is meant by "Varying parameters" as a modeling approach?

Response: It means we can use estimated parameters to represent processes that we didn't understand clearly, this method is introduced in detail by Luo and Schuur (2020). We will clarify this concept in the new version.

L138: What is meant by the leaf photosynthetic efficiency, and how is this connected to foliar C:N ratios? Also, formula (1) does not define LPE but SNvcmax. Are they the same? If not, what is SNvcmax? Also, what are the "estimated and defined C:N ratios of foliage"? How are they defined?

Response: Thank you for pointing this, here is a typo, LPE should be the same as SNvcmax here, the defined C:N ratios of foliage is the foliage C:N ratio estimated at the ambient treatment, and the estimated C:N ratios of foliage is the foliage C:N ratio estimated at different N addition treatments by data assimilation. We will modify these.

L186: The term C exit rates are introduced, but the reader is left alone with figuring out what these are. Therefore, they should be listed right away.

Response: Agreed, we will add the explanation when we revise it.

Results: Section 3.3 nicely guides the reader to the subplots in the figures to describe the results. This is missing in Section 3.2, which makes it difficult to understand.

Response: Thanks for your suggestion, we will make it clearer when revising it.

L229: What is the definition of a "well-constrained parameter"? There is no reference to any Figure or metric.

Response: Sorry for the confusion, we will add a description. The distributions of parameters usually have three types, bell shape, edge hitting, and flat, we regard the parameters of bell shape as well constrained (Huang *et al.*, 2021; Shi et al., 2016).

L370: What is the definition of significance to assess whether the estimated model parameters were significantly different?

Response: We selected 100 sets of parameter values from the posterior probability density function, and used one-way ANOVA to test the differences between different parameters with the significance of 95%.

L377: The sentence says that models with different structures should show similar dynamics, which is a confusing statement. Should the sentence rather state that the same

model applied to the same ecosystem across different N treatments should show similar responses?

Response: Here we want to illustrate that the carbon dynamics simulated by different models should be similar if they both do data assimilation with the same observations. We will revise these sentences to make them clearer.

L396: Unclear conceptualisation of N limitation. What is meant by "the decrease of the simulation"?

Response: Here we want to compare the results of C pool sizes and fluxes (e.g., GPP, leaf, soil in Figure 5 and Figure 6) estimated by three model and parameter combinations (Table 1). The first combination is the C-only model with the parameters estimated by the C-only model. The second combination is the C-N coupled model with the parameters estimated by the C-N coupled model. The third combination is the C-N model with the parameters estimated by the C-only model (Table 1). In this study, we assumed that the parameters estimated by the C-only model contained the N limitation information (effect). When we add an N module to the C-only model and use data assimilation to re-estimate parameters, the N limitation effects will be transformed from the C-related parameters to the N module in the C-N coupled model. Hence, the C pool sizes simulated by combination3 will be smaller than combination1 and combination2. These results were also shown in Figure 5 and Figure 6 in our manuscript.

Table 1 the Explanation of different model structures and parameter combinations

|              | Model structure | C-related parameters estimated from | N-related parameters estimated from |
| --- | --- | --- | --- |
| Combination1 | C-only | C-only model | — |
| Combination2 | C-N coupled | C-N coupled model | C-N coupled model |
| Combination3 | C-N coupled | C-only model | C-N coupled model |

L420: Detailed explanation is needed because it is not obvious how C exit rates cause a unimodal response in C pools and how a unimodal response provides information on nitrogen limitation.

Response: Agreed, to avoid confusion, we will delete this description.

The quality of language limits readability. The final version should be proofread by a native speaker to resolve semantic issues. Sentences that were difficult to read are on lines 45 (predicting C cycle), 46 (which remain), 63 (because-clause refers to no other sentence), 138 (sentence has no verb), 145 (The use of "similarly" is inappropriate), 158 ("no" should be "not"), 308 (sentence has no verb), 362 (unclear to what "their" refers to), 364 (unclear to what "those variables" refers to)

Response: Thank you for pointing out the problems. We have asked a native English speaker to polish the writing.

Formatting should be improved on lines 198, 203, 215, 218, 219, and 267.

Response: Thank you for pointing out the problems. We will revise them accordingly.

Kind regards on behalf of all co-authors,

Song Wang, Carlos A. Sierra, Yiqi Luo, Jinsong Wang, Weinan Chen, Yahai Zhang, Aizhong Ye, Shuli Niu

**Reference**

Bloom, A. A., J.-F. Exbrayat, I. R. Van Der Velde, L. Feng, and M. J. P. o. t. N. A. o. S. Williams. 2016. The decadal state of the terrestrial carbon cycle: Global retrievals of terrestrial carbon allocation, pools, and residence times, 113(5), 1285-1290.

Du, E., C. Terrer, A. F. A. Pellegrini, A. Ahlstrom, C. J. van Lissa, X. Zhao, et al. 2020. Global patterns of terrestrial nitrogen and phosphorus limitation, Nat Geosci, 13(3), 221-+.

Gerber, S., L. O. Hedin, M. Oppenheimer, S. W. Pacala, and E. Shevliakova. 2010. Nitrogen cycling and feedbacks in a global dynamic land model, Global Biogeochem Cy, 24.

Guo, X., Q. Gao, M. Yuan, G. Wang, X. Zhou, J. Feng, et al. 2020. Gene-informed decomposition model predicts lower soil carbon loss due to persistent microbial adaptation to warming, Nat Commun, 11(1), 4897.

Huang, X., D. Lu, D. M. Ricciuto, P. J. Hanson, A. D. Richardson, X. Lu, et al. 2021. A model-independent data assimilation (MIDA) module and its applications in ecology, 14(8), 5217-5238.

Liang, J., J. Xia, Z. Shi, L. Jiang, S. Ma, X. Lu, et al. 2018. Biotic responses buffer warming-induced soil organic carbon loss in Arctic tundra, Glob Chang Biol, 24(10), 4946-4959.

Luo, Y., and E. Schuur. 2020. Model parameterization to represent processes at unresolved scales and changing properties of evolving systems, Global Change Biol, 26(3).

Shi, Z., Y. H. Yang, X. H. Zhou, E. S. Weng, A. C. Finzi, and Y. Q. Luo. 2016. Inverse analysis of coupled carbon-nitrogen cycles against multiple datasets at ambient and elevated CO2, J Plant Ecol, 9(3), 285-295.

Yang, H., P. Ciais, Y. L. Wang, Y. Y. Huang, J. P. Wigneron, A. Bastos, et al. 2021. Variations of carbon allocation and turnover time across tropical forests, Global Ecol Biogeogr, 30(6), 1271-1285.

Zaehle, S., and A. D. Friend. 2010. Carbon and nitrogen cycle dynamics in the O-CN land surface model: 1. Model description, site-scale evaluation, and sensitivity to parameter estimates, Global Biogeochem Cy, 24(1), n/a-n/a.

---

## Author Comment (AC2)

**Responses to comments of Reviewer #2:**

We greatly appreciate the Referee #2's valuable comments, which we have used to substantially improve our manuscript. We have carefully considered the constructive comments and suggestions from the reviewers, and we provide answers to them here accordingly. Reviewer's comments are in black, our replies are in blue.

Wang et al. aim to retrieve N limitation information via data model fusion using a C-only model and coupled C-N model. They applied this approach to a field N enrichment experiment at an alpine meadow in the Qinghai-Tibet Plateau. The topic of nutrient limitation is of increasing importance in view of their role in constraining future land C sink in response to rising air CO2. I agree with most comments by Reviewer 1 and I have several additional comments that may further help to improve this work.

First, the term of nitrogen limitation needs to be clearly defined. Do you mean N limitation to plants, microbes or both? We don't usually say "ecosystem N limitation". Please also define "N limitation information" and clearly show how this is quantified in this study. Additionally, this manuscript seems to set up a background that N limitation occurs everywhere (L35-44). It would be helpful to provide an update of this view and mention that P instead of N is limiting in many tropical and subtropical ecosystems. The limitation by other nutrients needs to be mentioned or discussed in this manuscript. Second, there are many undefined terms and missing information in this manuscript (see specific comments). This hinders an in-depth evaluation of this work. Moreover, model structure and data assimilation are described in the method section but it is unclear how the two questions of this study were addressed (L91-93).

Response: Thank you for your comments. We will improve the description of the nitrogen limitation concept in the revised manuscript. In the context of this manuscript, the term of nitrogen limitation basically refers to limitation in plant primary production. We agree in that P and other nutrients limit process rates more strongly in other ecosystems. We will modify the manuscript accordingly.

For the detailed information about data assimilation methods, we will add them in the revised manuscript. We will also answer the questions more straightforward in the results and discussion parts.

Specific comments

L17-18 & 31: Exactly, how to provide guidance for policy making or ecosystem management?
Response: Sorry for the confusion, to avoid any potential misunderstandings, we will delete this description in the revised manuscript.

L24: Explain "carbon exit rates"

Response: "carbon exit rates" is the proportion of carbon outflow from a carbon pool at a time unit, these parameters represent the senescence rate of leaf and root, the decomposition rates of litter, and the decay rates of soil pools. We will add the explanation when we revise our manuscript.

L41-43: P limitation is also important but fully ignored.

Response: Agreed, we will add the P limitation effect when we revise the manuscript.

L45-48: The best way to address N limitation issues in earth system models is better understanding and representation of various N cycling processes (especially biological N fixation).

Response: Yes, we totally agree and will delete discussions on Earth System Models in the revised manuscript.

L45-55: Uncertainties are discussed for these methods. I would expect a description of the advantage of the data model fusion approach.

Response: Agreed, we will add the description of the data model fusion approach when we revise the manuscript.

L68: Explain "ecological information"

Response: "ecological information" refers to the information about ecological processes, such as the ecosystem functions, ecosystem services, nutrient limitation and so on. To avoid misunderstandings, we will change it to "information about ecological processes" and explain it when revising the manuscript.

L72-74: Any advantage (e.g., more accuracy) in comparison to other approaches(45-55)?

Response: Yes. We highlight two main advantages of our method to retrieve nutrient limitation. On the one hand, by comparing the parameters shift and pools size, we can more easily evaluate the N limitation effect on any specific C cycle process of an ecosystem. On the other hand, as long as some basic C and N variables are available, we can use them to evaluate the N limitation anywhere by a C-only model and the coupled C-N model with data assimilation technique.

L96-103: Provide additional information such as whether plant growth is N limited in the studied meadow and the level of N deposition.

Response: Agreed, we will add the information.

L139: The equation is not clear.

Response: Sorry for the typo, LPE is the same as SNvcmax here. We will correct it.

L227: Make sure that the GECO model simulated C pools of microbes.

Response: In this study, the microbial pool is used to constraint the fast SOM by a mapping vector which was introduced at 207-213. We will remove "microbe" in the new version.

L231-233: Again, explain "C exit rate"

Response: "carbon exit rates" is the proportion of carbon outflow from a carbon pool at a time unit, these parameters represent the senescence rate of leaf and root, the decomposition rates of litter, and the decay rates of soil pools. We will add the explanation when we revise our manuscript.

Figure 3: Does it mean C-N model overestimate LPE or C-only model underestimate LPE?
Response: It is likely that C-N model overestimated LPE when the observation was N limited. Because it needs to be multiplied by a nitrogen limiting scalar to match the observations, and the nitrogen limiting scalar should be less than 1 when the observation was N limited.

Figure 4: Please add titles for each axis

Response: Agreed. We will add.

Figure 5&6: It would be good to include the field observed data in comparison to the modelled results.

Response: Agreed, we will add the observed data to compare with the model results.

Figures 3-6: Any significant differences between modeling results?

Response: We used one-way ANOVA to test the differences between on different parameters and C pools and fluxes, we will add the significance in the revised figures.

L282-293: Not clear how N limitation was quantified and compared here. Do you mean the strength of N limitation changes with different levels of N additions for the same meadow?

Response: Yes, our basic assumption is that as nitrogen addition increases, the nitrogen limitation effect on the productivity and ecosystem carbon storage decreases.

Kind regards on behalf of all co-authors,

Song Wang, Carlos A. Sierra, Yiqi Luo, Jinsong Wang, Weinan Chen, Yahai Zhang, Aizhong Ye, Shuli Niu